# Bi-Phase NiCo$_2$S$_4$-NiS$_2$/CFP Nanocomposites as a Highly Active Catalyst for Oxygen Evolution Reaction

**Jintang Li *** , **Yongji Xia, Xianrui Luo, Tianle Mao, Zhenjia Wang, Zheyu Hong and Guanghui Yue ***

College of Materials, Xiamen University, Xiamen 361005, China
* Correspondence: leejt@xmu.edu.cn (J.L.); yuegh@xmu.edu.cn (G.Y.)

**Abstract:** Pursuing oxygen evolution reaction (OER) catalysts with high activity and stability is attracting many researchers. Here, we first designed and synthesized a biphasic three-dimensional structured catalyst of nickel–cobalt sulfide NiCo$_2$S$_4$-NiS$_2$/CFP, which is a nickel–cobalt sulfide composite decorated on carbon fiber paper (CFP) by a one-step hydrothermal method. It exhibited an overpotential of about 165 mV at a current density of 10 mA cm$^{-2}$ and a small Tafel slope of 81.54 mV dec$^{-1}$. Furthermore, the long-term stability of the NiCo$_2$S$_4$-NiS$_2$/CFP nanocomposite was 90.0% of the initial current density even after 12 h. The excellent catalytic performance of the NiCo$_2$S$_4$-NiS$_2$/CFP nanocomposite can be attributed to several aspects. Firstly, the petal-like morphology of the NiCo$_2$S$_4$-NiS$_2$ nanocomposite exposes more active sites. Secondly, the stability of the composite catalyst is significantly enhanced by its firm anchoring on the CFP. Thirdly, the catalytic performance was significantly improved by the addition of mixed valence of Ni or Co on the {111} plane of spinel NiCo$_2$S$_4$. Finally, the three-dimensional CFP substrate provides an efficient pathway and a stable integrated structure for the transmission of electrons and ions. Our one-step hydrothermal synthesis method provides a simple and economical way to obtain high-performance and robust OER catalysts.

**Keywords:** nickel–cobalt sulfide; oxygen evolution reaction; hydrothermal; electrocatalyst

## 1. Introduction

With today's society facing a severe fossil fuel energy crisis and environmental pollution, clean and sustainable energy sources are becoming a necessity for a highly civilized modern human society [1]. Therefore, searching for and developing new renewable energy sources is an urgent task at the moment [2–4]. The hydrogen energy which could be produced by water splitting [5,6] attracted the attention of many researchers because it is clean and inexpensive. However, owing to the kinetic sluggishness which results from four protons coupled with four-electron transfer in acidic and basic media, the efficiency of water splitting is greatly hindered by the half reaction oxygen evolution reaction (OER) [7–9]. RuO$_2$ and IrO$_2$ as the typical precious metal oxide catalysts are indicated as the state-of-the-art and best electrocatalysts for the OER, but the large-scale applications of those catalysts were blocked by their rareness and expensiveness [10,11]. Therefore, it is highly urgent and imperative to develop new and cheap electrocatalysts for OER with good electrochemical performance [12]. There are several ways to reduce the cost and improve the performance of catalysts, such as using metal doping [13], designing heterostructures [14] or deriving them from MOFs [15], which have been widely explored.

Recently, transition metal sulfides, which are abundant in the Earth's crust, have been widely investigated as potential OER catalysts due to their good electrical conductivity and high electrochemical activity [16–18]. However, some disadvantages of transition metal sulfides still limit their practical applications, such as scarce catalytic active sites, poor electron transport, low electrolyte contact efficiency and poor stability [19]. Many researchers have found that octahedral spinel structures contain more active sites than other structures, and thus are more likely to exhibit efficient catalytic activity [20–22]. It

is well known that $NiCo_2S_4$, a typical material with octahedral spinel structure, always has good catalytic properties [23,24]. A number of researchers have reported single-phase $NiCo_2S_4$ [25–27], but bi-phase $NiCo_2S_4$ catalysts are rarely mentioned.

Herein, a bi-phase $NiCo_2S_4$-$NiS_2$ decorated on the surface of carbon fiber paper (CFP) was synthesized via a simple one-step hydrothermal method, and the as-synthesized nanocomposites were used as the catalysts for OER. Based on the high conductivity of electron/ion, larger specific surface areas, plentiful transfer channels of mass and excellent mechanical strength [28], CFP not only worked as an excellent conductive substrate but also provided a support 3D structure for catalyst anchoring [29]. The active material is grown directly on the CFP to form a self-supporting catalyst, which avoids the loss of active sites and the decrease in electron/ion transportation with additional binder of Nafion solution and additives [30,31]. A one-step hydrothermal synthesis method of catalysts is more efficient and cost effective [16,32]. As a conclusion, the electrocatalytic performance would be improved with the exposure of more active sites which could be realized on the $NiCo_2S_4$-$NiS_2$/CFP catalyst. Here, the as-prepared $NiCo_2S_4$-$NiS_2$/CFP catalyst indicated a lower overpotential of 165 mV with a current density of 10 mA cm$^{-2}$ for OER, a small Tafel slope of 81.54 mV dec$^{-1}$ and a long-term stability of about 90.0% of its initial current density after 12 h.

## 2. Experimental Section

### 2.1. Reagents

All chemicals used in experiments were analytical reagents without further treatments. CFP was purchased from Dongli (Tianjin, China) Investment Co., Ltd. Cobalt nitrate hexahydrate ($Co(NO_3)_2 \cdot 6H_2O$), nickel acetate tetrahydrate ($Ni(CH_3COO)_2 \cdot 4H_2O$), potassium hydroxide, nitric acid and thiourea were bought from Xilong Science Co., Ltd. (Chengdu, China) Anhydrous ethanol, IrO2 powder and hydrochloric acid were bought from China Pharmaceutical Group Chemical Reagents Co., Ltd. (Shanghai, China) The dispersant sodium dodecyl sulfate (SDS) was supplied by Shantou Dahao Fine Chemicals Co., Ltd.

### 2.2. Treatment of CFP

First, a piece of CFP was cut into pieces with an area of $1.0 \times 1.0$ cm$^2$. Second, the CFP sheets were immersed in aqua regia solution (8 mL 65% $HNO_3$ mixed with 24 mL 37% HCl) and kept in a fume hood at room temperature for 6 h [26] to get rid of metals which remained on the surface of the CFP, to form a functionalized and hydrophilic surface. Finally, processed CFP sheets were washed with distilled water thoroughly to obtain oxidized carbon fiber paper (OCFP).

### 2.3. Synthesis of $NiCo_2S_4$-$NiS_2$/CFP, $Ni_3S_4$-$NiS_2$/CFP and $Co_2S$/CFP

All samples were synthesized by a hydrothermal method as follows and $Ni_3S_4$-$NiS_2$/CFP and $Co_2S$/CFP were also prepared in the same way as a reference group. Firstly, all solutions were prepared as shown for each solution listed in Table 1. Then, the solutions were transferred into a 100 mL Teflon-lined stainless-steel autoclave after being vigorously stirred for about half an hour, respectively. The autoclaves were kept at 220 °C for 12 h. After the autoclaves cooled to room temperature, the residues were collected with centrifugation and washed by anhydrous ethanol and deionized water three times, respectively. These three samples were dried in a vacuum desiccator at 80 °C for 12 h and denoted as $NiCo_2S_4$-$NiS_2$/CFP, $Ni_3S_4$-$NiS_2$/CFP and $Co_2S$/CFP. The average mass loading of $NiCo_2S_4$-$NiS_2$/CFP, $Ni_3S_4$-$NiS_2$/CFP and $Co_2S$/CFP on substrates was 0.81 mg cm$^{-2}$, 0.75 mg cm$^{-2}$ and 0.73 mg cm$^{-2}$, respectively.

**Table 1.** The synthesis of four kinds of sample.

| Sample | Thiourea (mol) | Co$^{2+}$ (mmol) | Ni$^{2+}$ (mmol) | SDS (mmol) | DI Water (mL) |
|---|---|---|---|---|---|
| NiCo$_2$S$_4$-NiS$_2$/CFP | 0.1 | 1.42 | 2.84 | 0.6 | 60 |
| Ni$_3$S$_4$-NiS$_2$/CFP | 0.1 | / | 4.26 | 0.6 | 60 |
| Co$_2$S/CFP | 0.1 | 4.26 | / | 0.6 | 60 |

*2.4. Physical Characterization*

The morphologies and compositions of samples were characterized by a field emission scanning electron microscope (FESEM, Hitachi SU-70, 10kV, Japan) equipped with an energy dispersive spectrometer (EDS, 20kV) analyzer. X-ray diffraction (XRD, Bruker, Germany) was implemented on a Bruker-AxsD8 X-ray diffractometer with a 2θ range from 10° to 90°. Rietveld refinement was accomplished by GSAS software. Transmission electron microscopy (TEM, Philips-FEI, Eindhoven, The Netherlands), high-resolution transmission electron microscopy (HRTEM), high-angle annular dark-field scanning transmission electron microscopy (HAADF-STEM) and corresponding energy dispersive spectroscopic (EDS) mapping analyses were all performed using a Talos F200s at 200 KV. X-ray photoelectron spectroscopy (XPS, USA) data were collected by using a PHI Quantum 2000 Scanning ESCA Microprobe with an Al X-ray source. The Raman measurement was performed in the XploRA microprobe Raman system (HORIBA, Kyoto, Japan).

*2.5. Electrochemical Testing*

All the electrochemical measurements were tested by employing an Autolab 302N in an electrochemistry workstation (Metrohm Autolab B.V., Utrecht, The Netherlands) at room temperature. The obtained CFP, NiCo$_2$S$_4$-NiS$_2$/CFP, Ni$_3$S$_4$-NiS$_2$/CFP and Co$_2$S/CFP were all cut into $0.5 \times 1$ cm$^2$ pieces and employed as working electrodes directly. The IrO$_2$ working electrode was prepared with the follow steps. Firstly, 5 mg IrO$_2$ powder was dispersed into 1 mL solution (anhydrous ethanol/5 wt.% Nafion, 19:1, *v/v*). Then, the solution was ultrasonicated for more than 30 min to obtain a homogeneous solution. Finally, 20 μL of the homogeneous solution was dropped onto the surface of glassy carbon (diameter is 5 mm) to obtain the working electrode (the catalyst loading was ~0.51 mg/cm$^2$). A mercuric oxide electrode and platinum (Pt) net were used as a reference electrode and counter electrode, respectively. The electrolyte was 1.0 M KOH (pH = 14). All measured potentials were calibrated with a reversible hydrogen electrode (RHE). The ultimate potential was calculated by the following equation: E (vs. RHE) = E (vs. Hg/HgO) + 0.098 V + 0.059 V·pH. Linear sweep voltammetry (LSV) was manipulated at a scan rate 5 mV/s, ranging from 1.0 V to 2.0 V (vs. RHE). Cyclic voltammetry (CV) tests were carried out with a scan rate of 10 mV/s, 20 mV/s, 40 mV/s, 60 mV/s, 80 mV/s and 100 mV/s, respectively, in the range of 1.0–1.1 V (vs. RHE). The working electrodes were scanned several times for CV measurements at a scan rate 5 mV/s, ranging from 1.0 V to 2.0 V (vs. RHE) until the curves were steady for LSV and CV testing. Electrochemical impedance spectra (EIS) were conducted at open circuit voltage (OCP, shown on the screen of the system) vs. RHE with 5 mV AC amplitude and frequency ranging from 0.1 Hz to 100 kHz. The stability of as-prepared catalysts was assessed with a chronoamperometry test.

**3. Results and Discussion**

*3.1. Synthesis and Characterization*

During the hydrothermal reaction, oxidized OCFP was reduced to CFP again because of using the reducing reagent thiourea, thus NiCo$_2$S$_4$-NiS$_2$ particles were decorated and anchored on the surface of CFP after the hydrothermal reaction at 220 °C to form the NiCo$_2$S$_4$-NiS$_2$/CFP nanocomposites (Figure S1). The simple synthetic process is illustrated as a schematic diagram in Figure 1.

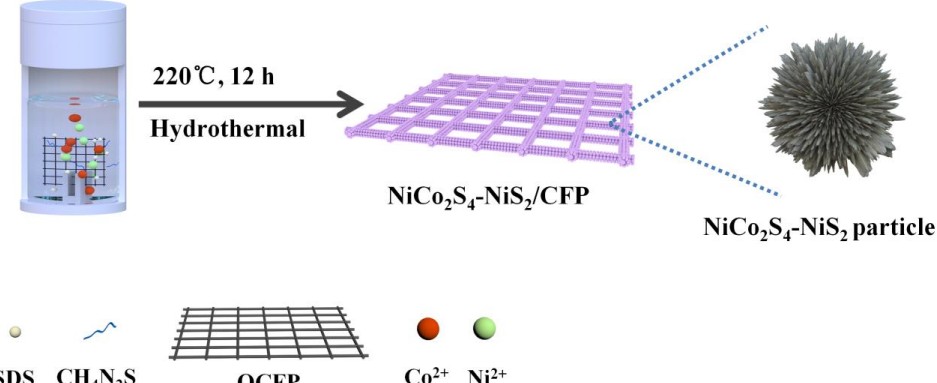

**Figure 1.** Schematic diagram showing the process for synthesizing NiCo$_2$S$_4$-NiS$_2$/CFP.

Figure 2 displays the morphology of the substrate and synthesized catalyst. Original commercial CFP was constituted by crossed microfibers with a diameter of approximately 7 μm and covered with carbon layers (Figure 2a), and a comparatively clear texture of the CFP can be found in Figure S2a,b. After being immersed in aqua regia for 6 h, CFP was oxidized into OCFP. It was detected that the surface of substrates had become smoother which means that metal impurities in CFP had been cleaned up without damaging the surface texture (Figure S2c,d). Petaloid-like NiCo$_2$S$_4$-NiS$_2$ with a diameter from 1 to 3 μm was formed by the aggregated NiCo$_2$S$_4$-NiS$_2$ nanowires, which were decorated and anchored on the surface of the CFP microfibers, as shown in Figure 2b,c. It can be found that the petaloid-like NiCo$_2$S$_4$-NiS$_2$ was distributed on the surface of the 3D skeleton of CFP uniformly, which makes the nanocomposites have a large specific surface and rough surface. The large specific surface, rough surface and the strong interaction may result in a good electrocatalytic performance of the as-synthesized NiCo$_2$S$_4$-NiS$_2$/CFP [33]. The morphologies of Ni$_3$S$_4$-NiS$_2$/CFP and CoS$_2$/CFP are shown in Figure S3a,b, respectively. The whole outline is the same as the sample of the NiCo$_2$S$_4$-NiS$_2$/CFP which is shown in Figure 2b. In Figure S3a,b, it can be seen that all Ni$_3$S$_4$-NiS$_2$ particles and CoS$_2$ particles are anchored on the surface of the CFP microfibers uniformly.

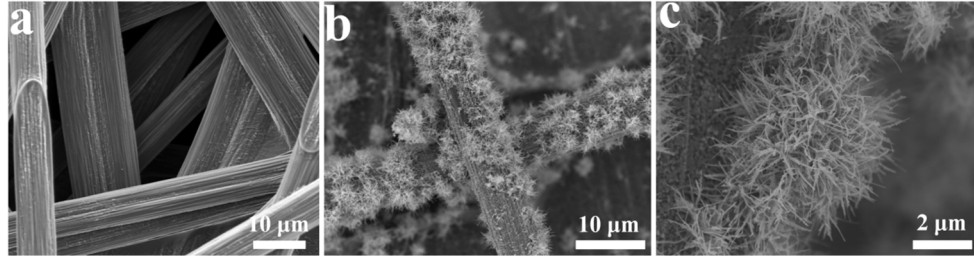

**Figure 2.** Morphologies of NiCo$_2$S$_4$-NiS$_2$/CFP. (**a**) SEM of CFP, (**b**,**c**) NiCo$_2$S$_4$-NiS$_2$/CFP.

The Rietveld refinement of XRD patterns of NiCo$_2$S$_4$-NiS$_2$ is shown in Figure 3. It suggests that the as-prepared electrocatalyst is composed of NiCo$_2$S$_4$ (JCPDS 20-0782) with a spinel structure (Fd-3m) and NiS$_2$ (JCPDS 73-0574). According to JCPDS 20-0782, the diffraction peak at 16.341° can be ascribed to the (111) plane of the NiCo$_2$S$_4$, which belongs to the {111} plane family of the spinel structure. Knözinger and Ratnasamy et al. [20,34] reported that the exposed octahedral cations in the {111} plane family of the spinel are more coordinated, resulting in better active catalytic performance. Furthermore, the weight percentages of NiCo$_2$S$_4$ phase and NiS$_2$ phase in the as-prepared catalyst are 87.34 wt% and 12.66 wt%, which were calculated by Rietveld structure refinement with GSAS software [35], as shown in Figure 3. The mole rate of NiCo$_2$S$_4$ phase and NiS$_2$ phase is 2.78:1, which means the spinel NiCo$_2$S$_4$ phase has a dominant position in the bi-phase NiCo$_2$S$_4$-NiS$_2$.

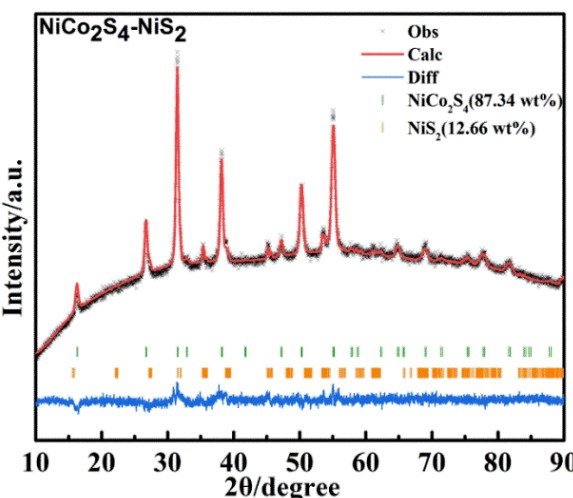

**Figure 3.** Rietveld refinement of XRD patterns of $NiCo_2S_4$-$NiS_2$.

The TEM image in Figure 4a shows that the petaloid-like $NiCo_2S_4$-$NiS_2$ is composed of many nanowires. The HRTEM image in the inset of Figure 4a shows that the petaloid-like $NiCo_2S_4$-$NiS_2$ nanocomposite is closely interlaced with $NiCo_2S_4$ and $NiS_2$, and these small particles are interlaced and stacked to form distinct lattice dislocations. The planar spacing of $NiCo_2S_4$ is 0.542 nm, which corresponds to the (111) lattice plane. The adjacent planar spacing is 0.328 nm which is attributed to the $NiS_2$ (1–11) lattice plane. From the selected area electron diffraction (SAED) image of Figure 4b, the polycrystalline electron diffraction rings of $NiCo_2S_4$-$NiS_2$ could be traced to the planes of $NiCo_2S_4$ (JCPDS 20-0782) and $NiS_2$ (JCPDS 73-0574), which agree with the XRD pattern of $NiCo_2S_4$-$NiS_2$ in Figure 3. As can be seen in Figure 4c–f, the EDS patterns show the presence of uniform dispersion of Ni, Co and S in the petal-like $NiCo_2S_4$-$NiS_2$ nanocomposites. The XRD patterns of $Ni_3S_4$-$NiS_2$ and $CoS_2$ are displayed in Figure S4a,b, and it can be seen that the $Ni_3S_4$-$NiS_2$ is composed of $Ni_3S_4$ (JCPDS 76-1813) and $NiS_2$ (JCPDS 88-1709). The HRTEM, SAED and the corresponding EDX elemental mapping of $Ni_3S_4$-$NiS_2$ and $CoS_2$ are shown in Figures S5 and S6, respectively.

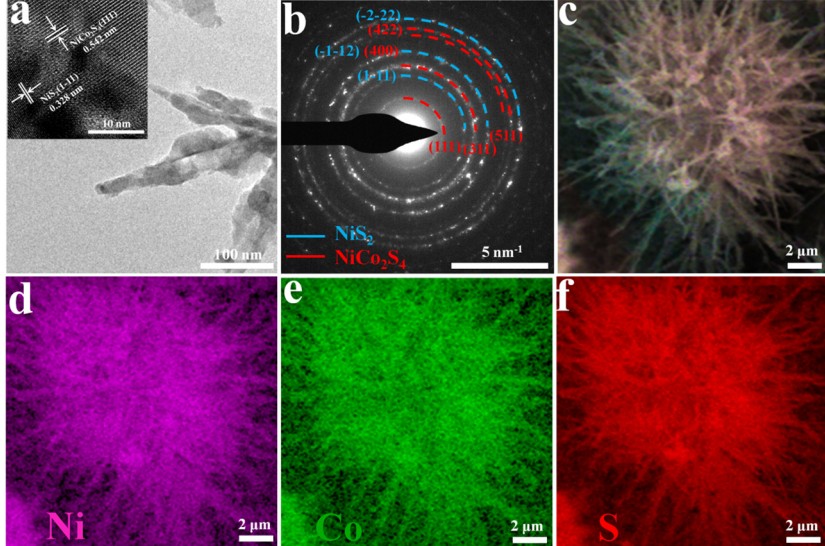

**Figure 4.** (**a**) The TEM image of $NiCo_2S_4$-$NiS_2$, the inset is HRTEM image of $NiCo_2S_4$-$NiS_2$. (**b**) Selected area electron diffraction (SAED) ring of $NiCo_2S_4$-$NiS_2$. (**c**–**f**) The area chosen for EDX elemental mapping of Ni, Co and S, respectively.

To further demonstrate that the $NiCo_2S_4$-$NiS_2$ particles were successfully immobilized on CFP, Raman spectroscopy was used to examine the $NiCo_2S_4$-$NiS_2$/CFP nanocomposites. The spectrum is shown in Figure S7, in which the strong peak which was located at around $1350$ cm$^{-1}$ corresponds to the D band, which implies a disorder or defect site of the sp$^2$ carbon atoms. Meanwhile, the other peak at around $1590$ cm$^{-1}$ can be attributed to the graphitic characteristic peak of the G band [36,37]. At the same time, the intensity of the D band/the intensity of the G band ($I_D/I_G$) indicates the degree of defects in carbon-based materials. $I_D/I_G$ values of CFP, OCFP and $NiCo_2S_4$-$NiS_2$/CFP are 0.247, 0.105 and 0.657, respectively. To maintain energy stability, $NiCo_2S_4$-$NiS_2$ nanoparticles prefer to anchor and assemble on the defect sites of CFP, which further leads to the increase in defects in $NiCo_2S_4$-$NiS_2$/CFP nanocomposites. All these consequences proved that the $NiCo_2S_4$-$NiS_2$ nanocomposites had been successfully anchored on CFP and self-assembled as flower petals to expose many active sites. Moreover, the small peak detected at around $1620$ cm$^{-1}$ (the red rectangle in Figure S7) in the CFP and $NiCo_2S_4$-$NiS_2$/CFP demonstrated OCFP was reduced to CFP during the hydrothermal reaction.

*3.2. Electrocatalytic Testing*

All the electrocatalytic performances of $NiCo_2S_4$-$NiS_2$/CFP nanocomposites were tested in 1M KOH solution. The $NiCo_2S_4$-$NiS_2$/CFP sample was used as a working electrode directly. The LSV curves are shown in Figure 5a, in which the $NiCo_2S_4$-$NiS_2$/CFP nanocomposites have the lowest overpotential of 165 mV at the current density of 10 mA cm$^{-2}$ in 1M KOH, compared to the $Ni_3S_4$-$NiS_2$/CFP nanocomposites (255 mV), the $CoS_2$/CFP nanocomposites (323 mV) and the commercial $IrO_2$ (368 mV). Electrocatalysis performance of $NiCo_2S_4$-$NiS_2$/CFP nanocomposites in this work is better than in other reported works which can be seen in Table S1. The OER activity of pure CFP was also tested for contrast, and a poor performance is revealed in Figure 5a. These results indicated that the high OER activity can be attributed to the transition metal sulfides. The large specific surface and rough surface provided by the petaloid spherules of $NiCo_2S_4$-$NiS_2$ nanocomposites have improved the electrocatalytic performance of the $NiCo_2S_4$-$NiS_2$/CFP. Compared with monometallic (Ni or Co) sulfide, the catalytic performance of the $NiCo_2S_4$-$NiS_2$/CFP nanocomposites could be improved by the synergistic effect of Ni and Co ions. Compared with $Ni_3S_4$-$NiS_2$/CFP, the catalytic performance of $NiCo_2S_4$-$NiS_2$/CFP could be boosted greatly with the existing spinel structure of $NiCo_2S_4$.

To study the catalytic kinetics of $NiCo_2S_4$-$NiS_2$/CFP, the Tafel slopes of all samples were determined from the LSV curves. According to the Tafel equation: $\eta$ = b log j + c, where $\eta$ is the potential (V vs. RHE), j is the current density (mA cm$^{-2}$), c is the constant, b is the Tafel slope. The Tafel slope b is an important parameter to understand the kinetic mechanism of OER, a smaller b means faster kinetics of OER. In Figure 5b, the $NiCo_2S_4$-$NiS_2$/CFP shows a Tafel slope of about 81.54 mV dec$^{-1}$, which is lower than for the CFP (211 mV dec$^{-1}$), $Ni_3S_4$-$NiS_2$/CFP (179 mV dec$^{-1}$) and $CoS_2$/CFP (218.35 mV dec$^{-1}$), with a value even lower than the noble-metal oxide $IrO_2$ (134.75 mV dec$^{-1}$). The ECSAs were estimated from $C_{DL}$ measurements using CV scans at different scan rates (Figure S8). Results showed that $NiCo_2S_4$-$NiS_2$/CFP possessed the biggest double-layer capacitance of 4.66 mF cm$^{-2}$ which is consistent with the large surface area of the petaloid-like morphology in SEM images. The Nyquist plots of electrocatalysts were obtained for the OER half-cell at open circuit voltage vs. RHE with 5 mV AC amplitude and the results are shown in Figure 5c. The semi-circle represents the charge transfer resistance ($R_{ct}$) of electrocatalysts. It shows that $R_{ct}$ of $NiCo_2S_4$-$NiS_2$/CFP is smaller than that of CFP, $Ni_3S_4$-$NiS_2$/CFP, $CoS_2$/CFP and $IrO_2$. The lower charge transfer resistance would prompt better electrocatalytic performance. The faster kinetic process of $NiCo_2S_4$-$NiS_2$/CFP can be attributed to the abundant proton transfer channels of CFP [28] and the mixed ions of $Ni^{2+}$, $Co^{2+}$ and $Co^{3+}$ in the $NiCo_2S_4$-$NiS_2$/CFP nanocomposites (Figure 6a,b).

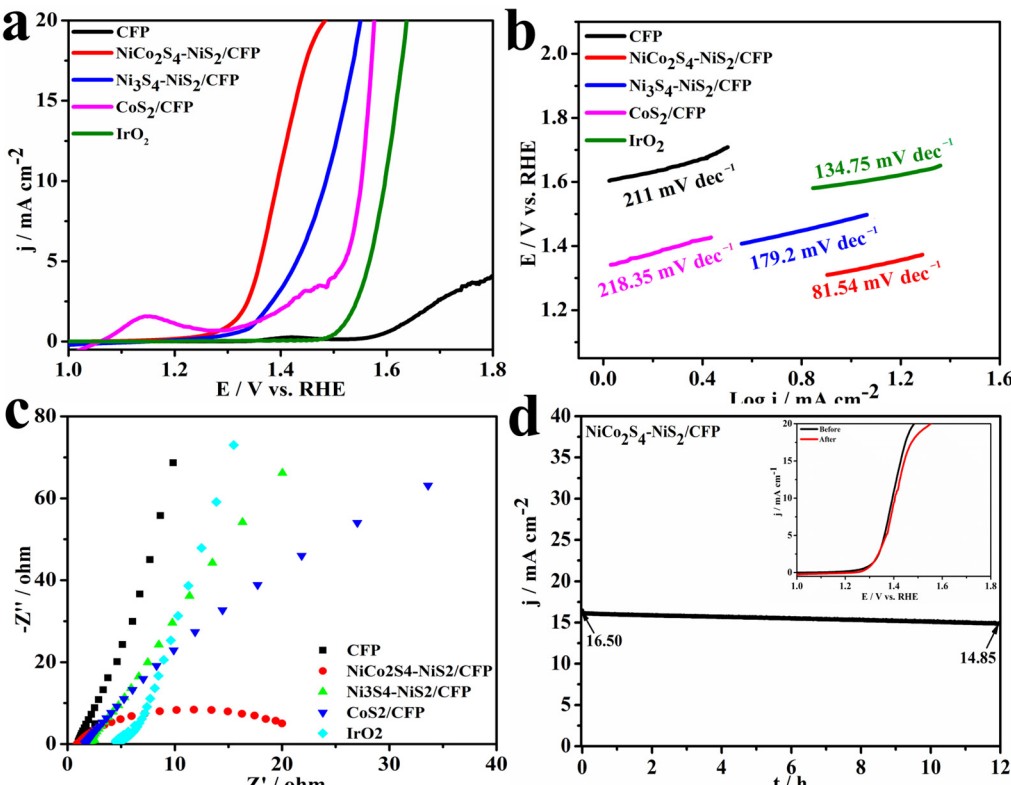

**Figure 5.** OER performance of $NiCo_2S_4$-$NiS_2$/CFP. (**a**) LSV of CFP, $NiCo_2S_4$-$NiS_2$/CFP, $Ni_3S_4$-$NiS_2$/CFP, $CoS_2$/CFP and commercial $IrO_2$. (**b**) Tafel slopes acquired from LSV in (**a**). (**c**) Nyquist plots of EIS. (**d**) The durability of $NiCo_2S_4$-$NiS_2$/CFP.

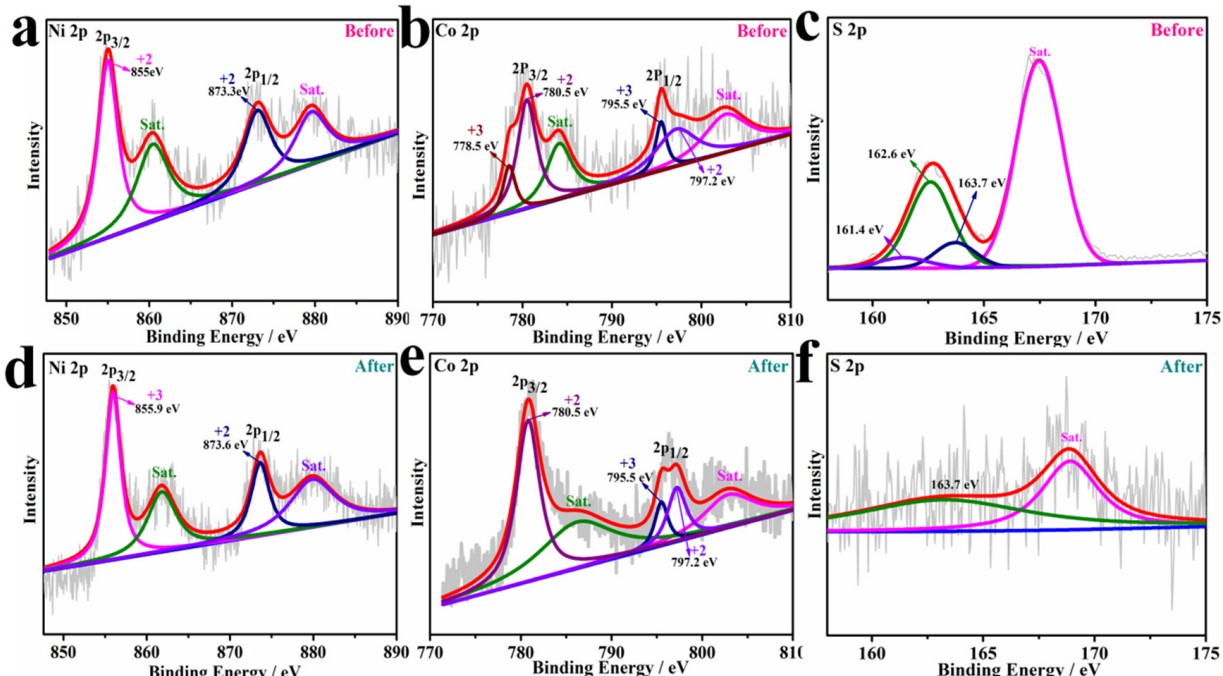

**Figure 6.** The XPS spectra of $NiCo_2S_4$-$NiS_2$/CFP before and after 12 h durability tests: (**a**,**d**) Ni 2p, (**b**,**e**) Co 2p and (**c**,**f**) S 2p.

Stability is an indispensable criterion for practical OER performance, so the long-term durability of the $NiCo_2S_4$-$NiS_2$/CFP sample was tested by chronoamperometric measurements at a constant potential of 1.4 V vs. RHE. As shown in Figure 5d, the

$NiCo_2S_4$-$NiS_2$/CFP nanocomposite retains an excellent durability of about 90.0% of the initial current density over a 12 h test. This can also be confirmed by the inset graphic in Figure 5d, showing the overpotential just increased a little bit after a 12 h durability test. Figure S9 reflects the morphology of $NiCo_2S_4$-$NiS_2$/CFP after a 12 h durability test, and the overall petaloid morphology is almost the same except the surface became smoother, resulting from NiCo2S4-NiS2 nanocomposites strongly anchored on CFP to form a strong binding between $NiCo_2S_4$-$NiS_2$ and CFP.

To further confirm the composition changes in as-prepared $NiCo_2S_4$-$NiS_2$/CFP before and after a 12 h durability test, XPS measurement was employed. Figure 6a,d show the Ni 2p spectra of $NiCo_2S_4$-$NiS_2$/CFP before and after a 12 h durability test, and it can be seen that the outlines of two Ni 2p are almost the same. The peaks located at 855 eV, 873.3 eV and 873.6 eV all correspond to the $Ni^{2+}$, the peaks located at 861.5 eV and 879.6 eV are labeled as the shake-up satellite peak [38,39] in Figure 6a,d. The peak located at 855.9 eV in Figure 6d is labeled as $Ni^{3+}$, which means some $Ni^{2+}$ was oxidized to $Ni^{3+}$ after the 12 h durability test. The Co 2p spectra before and after the 12 h durability test are displayed in Figure 6b,e, in which the peaks at 778.5 eV and 795.5 eV are ascribed to $Co^{3+}$ and the peaks at 780.5 eV and 797.2 eV are ascribed to $Co^{2+}$ [24,40], meaning that Co ions appeared with mixed valence in the spinel structure $NiCo_2S_4$ of $Ni_3S_4$-$NiS_2$/CFP. Meanwhile, the peaks at 803 eV and 784.5 eV are ascribed to two satellite peaks [41]. Compared with Figure 6b, the peak located at 778.5 eV (see Figure 6e) disappeared, which means some $Co^{3+}$ was reduced to $Co^{2+}$ after the 12 h durability test. It is assumed that the excellent stability can be attributed to the spinel-structured $Co^{3+}$ [17], and the 10% decrease in current density over the 12 h test could be ascribed to the reduction of $Co^{3+}$ in $NiCo_2S_4$. A distinct change in S is observed in Figure 6c,f. Before the 12 h durability test, the peaks situated at 163.7 eV and 161.4 eV could be attributed to the characteristic peak of metal sulfide bonds and $S^{2-}$ with low coordination, respectively [42,43]. The peak situated at 162.6 eV is attributed to bridging $S_2^{2-}$ [44], which suggests that non-saturated S atoms existed in the Ni–S areas in the $NiS_2$ phase, and it is assumed that the presence of bridging $S_2^{2-}$ may facilitate the protons' reduction and accelerate the electron transportation in the process of OER [17]. The peak situated at 167.4 eV is a satellite peak [45]. After the 12 h durability test, only a $S^{2-}$ peak located at 163.7 eV and a satellite peak at 168.9 eV could be detected, as the peaks of $S^{2-}$ and $S_2^{2-}$ situated at 161.4 eV and 162.6 eV (see Figure 6c) disappeared. Those results revealed that $S_2^{2-}$ and some $S^{2-}$ were oxidized into hydroxide within the OER process [46]. This could be confirmed with the O XPS spectra and the EDS results of the catalyst before and after the 12 h durability test (see Figures S10 and S11). In Figure S10, the oxygen vacancy peaks disappear after the 12 h durability test, the peaks at 530.4 eV and 531.5 eV could be attributed to the hydroxyl-metal (Co-OH, Ni-OH) and the peak at 532.2 eV could be attributed to adsorbed oxygen [46,47]. The changes in the S and O atoms are revealed by Figure S11, in which the decrease in S and increase in O indicate that part of the $S_2^{2-}$ and $S^{2-}$ had been oxidized to form the hydroxyl, which also corroborates the XPS results.

Compared with Ni 2p spectrum of $Ni_3S_4$-$NiS_2$/CFP shown in Figure S12a and Co 2p spectrum of $CoS_2$/CFP shown in Figure S13a, the mixed valences of $Ni^{2+}$, $Co^{2+}$ and $Co^{3+}$ existed in the catalyst $NiCo_2S_4$-$NiS_2$/CFP nanocomposites. The mixed metal valences can provide donor–acceptor chemisorption sites during the reversible adsorption of oxygen and then achieve relatively high electronic conductivity through valence electron hopping between cations [24,48].

The electrochemically effective surface area (ECSA) is also another parameter to evaluate the OER catalytic performance. It originated from double-layer capacitance ($C_{dl}$) on the catalyst surface, which satisfies the following equation: ECSA = $C_{dl}/C_s$ [49,50], while the value of $C_S$ in 1 M KOH is 0.04 mF cm$^{-2}$. $C_{dl}$ was calculated by cyclic voltammetry (CV) measured at scan rates from 10 to 100 mV/s. To obtain the currents related only to double-layer capacitance, scan rate-dependent CV was measured between 1.00 and 1.10 V (vs. RHE), where the redox processes do not occur (as shown in Figure 7). The $\triangle j = j_a - j_c$,

$j_a$ and $j_c$ were taken at 1.05 V vs. RHE. From the fitted linear slope, we calculated the $C_{dl}$ which is half of the slope [51]. Figure 7a reflects that the fitted linear slope values ranked from high to low are $NiCo_2S_4$-$NiS_2$/CFP > $IrO_2$ > $Ni_3S_4$-$NiS_2$/CFP > $CoS_2$/CFP. This result suggested that the ECSA of the as-prepared catalyst $NiCo_2S_4$-$NiS_2$/CFP is larger than that of the single metal (Ni and Co) sulfides and $IrO_2$, and the catalyst with larger ECSA could produce more ion-accessible sites and exhibit better electrocatalytic activity.

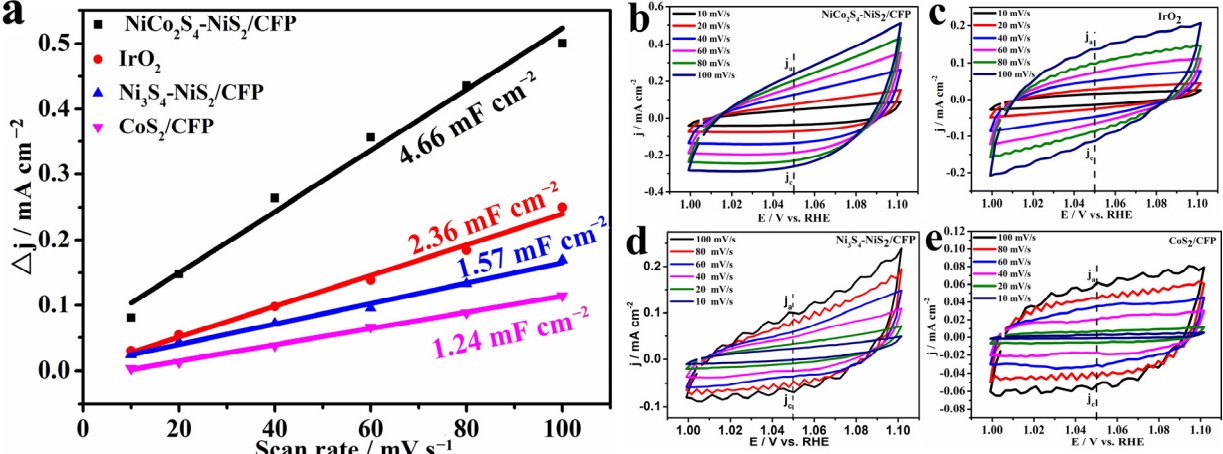

**Figure 7.** Electrochemical double-layer capacitances. (**a**) Linear fitting of $\triangle j$ at 0.10 V vs. RHE as a function of the scan rate. (**b**) $NiCo_2S_4$-$NiS_2$/CFP. (**c**) $IrO_2$. (**d**) $Ni_3S_4$-$NiS_2$/CFP. (**e**) $CoS_2$/CFP.

## 4. Conclusions

Here, petaloid-structured bi-phase bimetal sulfide catalyst $NiCo_2S_4$-$NiS_2$/CFP nanocomposites were synthesized by a simple hydrothermal method successfully. Compared with the monometallic (Ni or Co) sulfide with CFP, $NiCo_2S_4$-$NiS_2$/CFP exhibits remarkable performance with a low overpotential of 165 mV at a current density of 10 mA cm$^{-2}$ in alkaline solution, and also a detectable Tafel slope of 81.54 mV dec$^{-1}$. Under similar conditions, the overpotential of the synthesized $NiCo_2S_4$-$NiS_2$/CFP is lower than that of most recently reported electrocatalysts for OER. Meanwhile, a retention rate of 90.0% could be obtained when the catalyst was tested for 12 h. The superior performance of the as-prepared $NiCo_2S_4$-$NiS_2$/CFP nanocomposites benefits from several factors, including the special petaloid morphology and 3D structure provided by the CFP substrate, the strong coupling effect between spinel $NiCo_2S_4$ and $NiS_2$, the mixed valence of Ni and Co and the exposed active sites on the {111} plane. Furthermore, this work offers a simple and economical way via a hydrothermal method to obtain efficient and robust spinel $NiCo_2S_4$ base catalysts for OER.

**Supplementary Materials:** The following supporting information can be downloaded at: https://www.mdpi.com/article/10.3390/coatings13020313/s1, Figure S1. The surface change after as prepared NiCo2S4-NiS2 anchored on the surface of CFP; Figure S2. Low-magnification SEM images of (a) CFP and (c) OCFP. High-magnification SEM images of (b) CFP and (d) OCFP; Figure S3. The morphologies of (a) Ni3S4-NiS2/CFP and (b) CoS2/CFP; Figure S4. XRD patterns of (a)Ni3S4-NiS2 and (b) CoS2; Figure S5. (a,b) HRTEM image of Ni3S4-NiS2. (c,d) Selected area electron diffraction (SAED) diffraction ring of Ni3S4-NiS2. (e,f) The area chosen to do EDX elemental mapping and the corresponding EDX elemental mapping images for Ni and S respectively; Figure S6. (a) The TEM image of CoS2. (b) HRTEM image of CoS2. (c,d) Selected area electron diffraction (SAED) diffraction ring of CoS2. (e,f) The area chosen to do EDX elemental mapping nd the corresponding EDX elemental mapping images for Co and S respectively; Figure S7. Raman spectra of CFP, OCFP and NiCo2S4-NiS2/CFP; Figure S8. Double-layer capacitances (a) and CV scans at different scan rates (b–e); Figure S9. The morphology of NiCo2S4-NiS2/CFP after 12 h durability test; Figure S10. The O XPS spectra of as-prepared catalyst before (a) and after (b)12 h durability test; Figure S11. The EDS results of NiCo2S4-NiS2/CFP (a) before 12 h durability test. (b) After 12 h durability test; Figure S12.

The XPS spectra of (a) Ni 2p, (b) S 2p in Ni3S4-NiS2/CFP; Figure S13. The XPS spectra of (a) Co 2p, (b) S 2p in CoS2/CFP; Table S1. Comparison of OER activity of NiCo2S4-NiS2/CFP with that of most reported nick-el-cobalt sulfide catalysts tested in alkaline solution [52–58].

**Author Contributions:** Conceptualization, writing—review and editing, J.L.; methodology, Y.X., X.L., and T.M.; formal analysis, Y.X., Z.W. and Z.H.; supervision and funding acquisition, G.Y. All authors have read and agreed to the published version of the manuscript.

**Funding:** This research was funded by the National Natural Science Foundation of China (Grant No. 51971184), the Natural Science Foundation of Fujian Province of China (No. 2021J01043).

**Institutional Review Board Statement:** Not applicable.

**Informed Consent Statement:** Not applicable.

**Data Availability Statement:** Not applicable.

**Conflicts of Interest:** The authors declare have no conflict of interest.

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
