# Peer review of "Bi-Phase NiCo2S4-NiS2/CFP Nanocomposites as a Highly Active Catalyst for Oxygen Evolution Reaction"

_coatings, doi:10.3390/coatings13020313_

Round 1
Reviewer 1 Report
1. “The introduction part looks too casual. The authors should rewrite the introduction part with some recent references related to this”.
2. “The authors should explain the advantage of the hydrothermal method compared to other techniques.
3. The authors discussed the structural properties of the prepared composites by XRD and Rietveld refinement of XRD patterns was carried out for the phase confirmation of the synthesized composites. The authors should calculate the crystallite size of the synthesized composites.
4. The authors also studied the surface properties of the synthesized composites using X-ray photoelectron spectroscopy (XPS). What is the B.E. of the standard C─C bond of the calibrated XPS instrument? The samples could be studied the synthesized composites without charge correction if all the samples were studied by using the Wagner plot with the modified Auger parameters (MAP). The authors didn’t discuss the charge correction of the synthesized before analyzing the XPS high-resolution data. The authors should give some references for the charge correction of the synthesized samples e.g. Surface, optical and photocatalytic properties of Rb doped ZnO nanoparticles; Optical limiting applications of resonating plasmonic Au nanoparticles in a dielectric glass medium; Plasmonic Au nanoparticles embedded in glass: Study of TOF-SIMS, XPS and its enhanced antimicrobial activities.
5. In the XPS spectrum of (a) Ni 2p (Fig.S11), the deconvolution is not correct. The authors should deconvolute it once again and then study the oxidation states of the Ni 2p spectrum.

Author Response
The following is a point-by-point response to the reviewer’s comments.
Responds to the reviewers’ comments:
Reviewer:
- “The introduction part looks too casual. The authors should rewrite the introduction part with some recent references related to this”.
Response: Thanks for your kind suggestions. We have carefully checked our manuscript several times and made some revisions at the introduction part. We also add several recent references. We are using “track changes” model in MS word.
- “The authors should explain the advantage of the hydrothermal method compared to other techniques.
Response: Thanks for your kind suggestions. The hydrothermal method is an efficient method which are widely used in inorganic material synthesis. Other synthesized method like solution synthesis always takes several days or weeks. Mechanical chemistry method is more efficient but could not obtain nanosized particles. We mentioned in the revised manuscript.
- The authors discussed the structural properties of the prepared composites by XRD and Rietveld refinement of XRD patterns was carried out for the phase confirmation of the synthesized composites. The authors should calculate the crystallite size of the synthesized composites.
Response: Thanks for your kind suggestions. We calculated the crystallite size from XRD data using the Scherrer equation.
Where K=0.9 (Scherrer constant), λ=0.15406 nm (copper target), β=FWHM(radians). The average crystallite size is 37.0897 nm using five peaks.
- The authors also studied the surface properties of the synthesized composites using X-ray photoelectron spectroscopy (XPS). What is the B.E. of the standard C─C bond of the calibrated XPS instrument? The samples could be studied the synthesized composites without charge correction if all the samples were studied by using the Wagner plot with the modified Auger parameters (MAP). The authors didn’t discuss the charge correction of the synthesized before analyzing the XPS high-resolution data. The authors should give some references for the charge correction of the synthesized samples e.g. Surface, optical and photocatalytic properties of Rb doped ZnO nanoparticles; Optical limiting applications of resonating plasmonic Au nanoparticles in a dielectric glass medium; Plasmonic Au nanoparticles embedded in glass: Study of TOF-SIMS, XPS and its enhanced antimicrobial activities.
Response: Thanks for your kind suggestions. The B.E. of the standard C-C bond of the calibrated XPS instrument is about 284.8 eV. We also did the charge neutralization according to some references.
- In the XPS spectrum of (a) Ni 2p (Fig.S11), the deconvolution is not correct. The authors should deconvolute it once again and then study the oxidation states of the Ni 2p spectrum.
Response: Thanks for your kind suggestions. We fixed it.
Reviewer 2 Report
Journal: Coatings
Type of manuscript: Article
Title: Bi-phase NiCo2S4-NiS2/CFP nanocomposites as a highly active catalyst for oxygen evolution reaction
Authors: Jintang Li, Yongji Xia, Xianrui Luo, Tianle Mao, Zhenjia Wang, zheyu Hong, Guanghui Yue
Reviewing report
The manuscript presents various data on oxygen evolution reaction (OER) using a series of nickel-cobalt sulfide composite catalysts for which carbon fiber paper served as immobilizing support with hydrothermal method as preparation technique. The authors based their structure-morphology characterization on the use of XRD associated with Rietvel refinement, Raman spectroscopy, various microscopy analyses (FTSEM, TEM, EDS mapping …) and surface analysis using XPS. The electrocatalytic performance of catalysts was evaluated using electrochemical measurements basically voltammetry (LSV and CV) and impedance spectra collection (EIS).
They concluded the superior performance of NiCo2S4- NiS2/CFP was exhibited by a current density of 10 mA cm-2 and a small Tafel slope of 81.54 9 mV dec-1, in addition to a retention rate of current density of 90%. The authors explained the performance of best catalyst by the extent of active surface defined by morphological control on the shape of catalyst particles, the strength of junction between Ni-Co composite and CFP support and to chemical interaction at the surface of the nanocomposite between mixed valences of Ni and Co in a cooperative work between what they supposed to be determining active sites in the OER process
This paper as presented cannot be considered for immediate publication in “Coatings” and needs to be thoroughly reviewed according to standards required by the journal before being considered to be published.
Because: - The English writing must be improved to reach the standard required for this journal. - There are no convincing evidences for the chemical state of Nickel and Cobalt during reaction and for the role of Ni/Co/S catalyst sites in the process. Further comparison must be made using in situ monitoring techniques. check separately monometallic spinel nanocomposite Ni and Co sulfides in the OER process; - Procedure to calculate number of active sites is missing; - The authors used XPS for decide about the distribution of various species on the surface, how did they insure the real configuration during the reaction?. It is difficult to base the determination of mixed valence species content on the surface using just deconvolution of XPS peaks. That is not a sufficient way. One approach to use may be probe molecules, this will allow to estimate more precisely the metal coverage; - Auger Parameters from XPS might be used for discuss the dispersion of metals species (shift observed to higher values with increasing metal content? ). Is this coherent with results obtained from XRD-SEM? - It is strongly recommended to perform kinetic measurement not only on the basis of Tafel equation, it is necessary to take into account the conversion in a differential regime of OER to be indicative about a measurable rate. Finally, - Your literature survey needs improvement; please expand and diversify it in order to conclude about the performance of your catalyst and its active species in OER operation In my opinion, OER as it is proposed in this work on these catalysts is interesting but not sufficiently elucidated.I encourage the authors of this paper to reinforce the content with more effective results to avoid speculative conclusions,
The paper must be substantially enriched and thoroughly reviewed before being considered to be published in Coatings.
Author Response
The following is a point-by-point response to the reviewer’s comments.
Responds to the reviewers’ comments:
Reviewer:
This paper as presented cannot be considered for immediate publication in “Coatings” and needs to be thoroughly reviewed according to standards required by the journal before being considered to be published.
Because: - The English writing must be improved to reach the standard required for this journal. - There are no convincing evidences for the chemical state of Nickel and Cobalt during reaction and for the role of Ni/Co/S catalyst sites in the process. Further comparison must be made using in situ monitoring techniques. check separately monometallic spinel nanocomposite Ni and Co sulfides in the OER process; - Procedure to calculate number of active sites is missing; - The authors used XPS for decide about the distribution of various species on the surface, how did they insure the real configuration during the reaction?. It is difficult to base the determination of mixed valence species content on the surface using just deconvolution of XPS peaks. That is not a sufficient way. One approach to use may be probe molecules, this will allow to estimate more precisely the metal coverage; - Auger Parameters from XPS might be used for discuss the dispersion of metals species (shift observed to higher values with increasing metal content? ). Is this coherent with results obtained from XRD-SEM? - It is strongly recommended to perform kinetic measurement not only on the basis of Tafel equation, it is necessary to take into account the conversion in a differential regime of OER to be indicative about a measurable rate. Finally, - Your literature survey needs improvement; please expand and diversify it in order to conclude about the performance of your catalyst and its active species in OER operation In my opinion, OER as it is proposed in this work on these catalysts is interesting but not sufficiently elucidated.
I encourage the authors of this paper to reinforce the content with more effective results to avoid speculative conclusions,
The paper must be substantially enriched and thoroughly reviewed before being considered to be published in Coatings.
- The English writing must be improved to reach the standard required for this journal.
Response: Thanks for your kind suggestions. We have carefully checked our manuscript several times and made some revisions at the introduction part. We are using “track changes” model in MS word for your convenience to check.
- There are no convincing evidences for the chemical state of Nickel and Cobalt during reaction and for the role of Ni/Co/S catalyst sites in the process. Further comparison must be made using in situ monitoring techniques. check separately monometallic spinel nanocomposite Ni and Co sulfides in the OER process;
Response: Thanks for your kind suggestions. There are some difficulties for in situ monitoring test since we are using hydrothermal method which is at high press and temperature. But your advice is very valuable, we will figure out new method in future research. Monometallic Ni or Co sulfides are not tending to crystallize in spinel structure. But we compared our composite catalyst with single phase NiCo2S4 or Ni3S4-NiS2/CFP in other references and showed better OER catalysis performance.
- Procedure to calculate number of active sites is missing;
Response: Thanks for your kind suggestions. We can’t exactly calculate the number of active sites. But our catalyst possessing large specific surface and rough surface may have more active sites. Also electrochemical tests showed the improved electrocatalytic performance of the NiCo2S4-NiS2/CFP.
- The authors used XPS for decide about the distribution of various species on the surface, how did they insure the real configuration during the reaction? It is difficult to base the determination of mixed valence species content on the surface using just deconvolution of XPS peaks. That is not a sufficient way. One approach to use may be probe molecules, this will allow to estimate more precisely the metal coverage;
Response: Thanks for your kind suggestions, you are right. It is difficult to determine the mixed valence species content on the surface using just deconvolution of XPS peaks. We used XRD diffraction data to calculate the mole rate of NiCo2S4 phase and NiS2 phase is 2.78:1. Using probe molecules may be a good way but our powder samples are difficult to do that.
- Auger Parameters from XPS might be used for discuss the dispersion of metals species (shift observed to higher values with increasing metal content? ). Is this coherent with results obtained from XRD-SEM?
Response: Thanks for your kind suggestions. We did the XPS measurement mainly to study the durability of the catalyst. We didn’t analysis the Auger parameters from XPS.
- It is strongly recommended to perform kinetic measurement not only on the basis of Tafel equation, it is necessary to take into account the conversion in a differential regime of OER to be indicative about a measurable rate.
Response: Thanks for your kind suggestions. We may use in situ monitoring test in our future research to know the exact kinetic of the catalysis process.
- Finally, Your literature survey needs improvement; please expand and diversify it in order to conclude about the performance of your catalyst and its active species in OER operation In my opinion, OER as it is proposed in this work on these catalysts is interesting but not sufficiently elucidated.
Response: Thanks for your kind suggestions. We renewed and added several recent references. We made change in the manuscript and using “track changes” model in MS word.
Reviewer 3 Report
The authors presented work on the Bi-phase NiCo2S4-NiS2/CFP nanocomposites as a highly active catalyst for oxygen evolution reaction. The manuscript is presented in well-arranged data order and written very well. The authors have carried out a systematic investigation. The results are interesting and well-supported by experimental/analytical evidence. Therefore, I would recommend this article for publication in the coatings after some minor revisions as advised in my comments below:
- The the introduction does not suit with the aim of the study. When generally introduce the energy storage and metal-air batteries, some highly relevant articles should be included in the manuscript: (https://doi.org/10.1016/j.surfin.2022.102410), (https://doi.org/10.1007/s10971-022-05961-3), (https://doi.org/10.1016/j.ceramint.2022.10.166) . There are many reports are available. What is the novelty of the current work? The authors need to clarify.
-the loading of the catalysts should be provided.
-“The large specific surface, rough surface provided by the petaloid spherules NiCo2S4-NiS2 nanocomposites have improved the electrocatalytic performance”. Please explain with proper reason.
-It might seem that morphology by SEM is several micrometres but from TEM, within the nm range.
-it is more interesting to explain the exact mechanism of the excellent OER performance for the composite, which is missing in the text. SEM shows that there is almost no interaction between the two components. why was the OER performance enhanced?
Author Response
Responds to the reviewers’ comments:
Reviewer:
- The the introduction does not suit with the aim of the study. When generally introduce the energy storage and metal-air batteries, some highly relevant articles should be included in the manuscript: (https://doi.org/10.1016/j.surfin.2022.102410), (https://doi.org/10.1007/s10971-022-05961-3), (https://doi.org/10.1016/j.ceramint.2022.10.166) . There are many reports are available. What is the novelty of the current work? The authors need to clarify.
Response: Thanks for your kind suggestions. We have carefully checked our manuscript several times and made some revisions at the introduction part. We also add several recent references including your mentioned references. We are using “track changes” model in MS word.
- -the loading of the catalysts should be provided.
Response: Thanks for your kind suggestions. The catalyst loading was ~0.51 mg/cm2.
- “The large specific surface, rough surface provided by the petaloid spherules NiCo2S4-NiS2 nanocomposites have improved the electrocatalytic performance”. Please explain with proper reason.
Response: Thanks for your kind suggestions. The rougher surface will expose more active catalysis sites according to other research. Compared to regular morphology NiCo2S4, the catalyst we synthesized was more efficient.
- It might seem that morphology by SEM is several micrometres but from TEM, within the nm range.
Response: Thanks for your kind asking. The diameter of nanocomposite particles is about 50 nm. The petaloid morphology is composed of much smaller particles which SEM did not have enough resolution to observe.
- it is more interesting to explain the exact mechanism of the excellent OER performance for the composite, which is missing in the text. SEM shows that there is almost no interaction between the two components. why was the OER performance enhanced?
Response: Thanks for your kind suggestions. It’s important to explain the mechanism of excellent OER performance for the composite. But it will need elaborate determination and theoretical calculation. We are working on it.
Reviewer 4 Report
Hydrogen is one of the most promising energy carriers that can be produced in a carbon neutral way by electrolysis utilizing renewable electricity only. Here the authors report the fabrication of NiCo based sulfide supported on carbon fiber paper. While the characterization of the as-received material is solid, the electrochemical data presented has serious flaws.
- EIS needs a proper model to be suitable as a characterization tool. The Nyquist plots shown are meaningless without proper analysis
- The used reference, IrO2 on carbon, is clearly not suitable as the electrode structure, ie IrO2 on a flat disc, is significantly different from the 3D structure provided by the CFP support
- The authors claim low overpotentials for their material but it is highly questionable to what extent O2 is formed. In fact Ni and Co based materials are known to have a redox wave in the potential region mentioned here, ie Ni2+ to Ni3+, which is completely ignored. The fact that at higher potentials the a plateau current is observed is fully in line with the speculation that only the oxidation of the electrode is observed. Another hint is the significant change in Tafel slope shown in the SI
- Related to the comment above, Tafel plots are not convincingly constructed
- It is arguable that the shown stability is simply caused by the tremendous surface area. Certainly the electrode after testing is not resembling the initial state as shown by SEM and XPS.
In short the manuscript is not recommended for publication
Author Response
Response: Thanks for your kind suggestions, It’s very constructive. We improved our manuscript according to your suggestion. Also, your advice will enhance our future research.
Comments and Suggestions for Authors
Hydrogen is one of the most promising energy carriers that can be produced in a carbon neutral way by electrolysis utilizing renewable electricity only. Here the authors report the fabrication of NiCo based sulfide supported on carbon fiber paper. While the characterization of the as-received material is solid, the electrochemical data presented has serious flaws.
- EIS needs a proper model to be suitable as a characterization tool. The Nyquist plots shown are meaningless without proper analysis.
Response: Thanks for your kind suggestions, It’s very valuable. It’s difficult to establish the proper model since we didn’t know the exact state of the electrocatalysis reaction about the synthesized catalyst. We tested the EIS and investigated the Nyquist plots to compare the average charge transfer resistance of different catalyst. It showed NiCo2S4-NiS2/CFP had the smallest resistance.
- The used reference, IrO2 on carbon, is clearly not suitable as the electrode structure, ie IrO2 on a flat disc, is significantly different from the 3D structure provided by the CFP support.
Response: Thanks for your kind suggestions, you are right. We can’t find a better reference similar to our 3D structure catalyst. Since IrO2 is a benchmark catalyst for general OER test, so we chose it too.
3.The authors claim low overpotentials for their material but it is highly questionable to what extent O2 is formed. In fact Ni and Co based materials are known to have a redox wave in the potential region mentioned here, ie Ni2+ to Ni3+, which is completely ignored. The fact that at higher potentials the a plateau current is observed is fully in line with the speculation that only the oxidation of the electrode is observed. Another hint is the significant change in Tafel slope shown in the SI Related to the comment above, Tafel plots are not convincingly constructed
It is arguable that the shown stability is simply caused by the tremendous surface area. Certainly the electrode after testing is not resembling the initial state as shown by SEM and XPS.
In short the manuscript is not recommended for publication
Response: Thanks for your kind suggestions, you are right. According to many references, catalysts surface will change during the electrochemistry reaction. The reaction intermediate state is irreversible. The Tafel plots were determined from the LSV curves according to the Tafel equation. The smaller Tafel slope means the faster kinetic of OER according to the references. We also improved our manuscript according to your suggestion, your advice will enhance our future research.
Round 2
Reviewer 2 Report
HELLO
Authors made some efforts to improve the paper, however for their future resaech development , especially concerning material chracterization and ORR kinetics, I recommend to the authors to reconsider their approach .
In my opinion, taking into account the substantial work done, this paper can be published in its present revised form
Author Response
Responds to the reviewers’ comments:
Reviewer:
Authors made some efforts to improve the paper, however for their future resaech development , especially concerning material chracterization and ORR kinetics, I recommend to the authors to reconsider their approach .
In my opinion, taking into account the substantial work done, this paper can be published in its present revised form
Response: Thanks for your kind suggestions and your comments are very valuable. We will make more efforts in material characterization and electrocatalysis kinetics research in our future work.
Reviewer 4 Report
This is the second time I am reviewing this article. Unfortunately, I don't see any improvement, specifically comment 3 has not been addressed, which could have been easily done by O2 quantification, further analysis of the current-potential relation (ie scanning to higher potentials), testing of the materials at more relevant current densities, well beyond 100 mA/cm2 and normalization by active rather than geometrical surface area. Thus I am still not recommending the article for publication
Author Response
Responds to the reviewers’ comments:
Reviewer:
This is the second time I am reviewing this article. Unfortunately, I don't see any improvement, specifically comment 3 has not been addressed, which could have been easily done by O2 quantification, further analysis of the current-potential relation (ie scanning to higher potentials), testing of the materials at more relevant current densities, well beyond 100 mA/cm2 and normalization by active rather than geometrical surface area. Thus I am still not recommending the article for publication
Response: Thanks for your professional suggestions, it’s very valuable. Your expertise in the electrocatalytic analysis is quite impressive. We characterized our materials according to some references (Chem. Mater., 2017, 9(1), 120-140; ChemCatChem,2022, 14, e202101280). But your suggestions have given us new instructions and inspiration to perform electrochemical analysis. There are so many works to be done and we will put more effort into this area.